# Mitigating Osmotic Stress and Enhancing Developmental Productivity Processes in Cotton through Integrative Use of Vermicompost and Cyanobacteria

**DOI:** 10.3390/plants12091872

**Published:** 2023-05-03

**Authors:** Khadiga Alharbi, Emad M. Hafez, Alaa El-Dein Omara, Hany S. Osman

**Affiliations:** 1Department of Biology, College of Science, Princess Nourah bint Abdulrahman University, Riyadh 11671, Saudi Arabia; kralharbi@pnu.edu.sa; 2Department of Agronomy, Faculty of Agriculture, Kafrelsheikh University, Kafr El-Sheikh 33516, Egypt; 3Department of Microbiology, Soils, Water Environment Research Institute, Agricultural Research Center, Giza 12112, Egypt; alaa.omara@yahoo.com; 4Department of Agricultural Botany, Faculty of Agriculture, Ain Shams University, Hadayek Shubra, Cairo 11241, Egypt

**Keywords:** cotton, vermicompost, *Spirulina*, blue-green algae, cyanobacteria, deficit irrigation, drought, salt-affected soil

## Abstract

There is an urgent demand for biostimulant amendments that can sustainably alleviate osmotic stress. However, limited information is available about the integrated application of vermicompost and a cyanobacteria extract on cotton plants. In 2020 and 2021, two field experiments were carried out in which twelve combinations of three irrigation intervals were employed every 14 days (Irrig.14), 21 days (Irrig.21), and 28 days (Irrig.28) along with four amendment treatments (a control, vermicompost, cyanobacteria extract, and combination of vermicompost + cyanobacteria extract) in salt-affected soil. The integrative use of vermicompost and a cyanobacteria extract resulted in an observed improvement in the physicochemical attributes; non-enzymatic antioxidants (free amino acids, proline, total soluble sugars, and phenolics); and antioxidant enzyme activities of catalase (CAT), superoxide dismutase (SOD), and peroxidase (POD) and a decrease in the levels of oxidative damage indicators (H_2_O_2_ and MDA). Significant augmentation in the content of chlorophyll *a* and *b*, carotenoid concentration, relative water content, stomatal conductance, and K^+^ was also observed. In conjunction with these findings, noticeable decreases in the content of Na^+^ and hydrogen peroxide (H_2_O_2_) and the degree of lipid peroxidation (MDA) proved the efficacy of this technique. Consequently, the highest cotton yield and productivity as well as fiber quality were achieved when vermicompost and a cyanobacteria extract were used together under increasing irrigation intervals in salt-affected soil. In conclusion, the integrated application of vermicompost and a cyanobacteria extract can be helpful for obtaining higher cotton productivity and fiber quality compared with the studied control and the individual applications of the vermicompost or the cyanobacteria extract under increasing irrigation intervals within salt-affected soil. Additionally, it can also help alleviate the harmful impact of these abiotic stresses.

## 1. Introduction

Cotton (*Gossypium barbadense* L.) is one of the most profitable crops in the world, especially in Egypt. It plays an essential part in the country’s agricultural and industrial development as it is one of the main global cash crops. Cotton (x = 2n = 26, or n = 13), as an annual crop, is mainly cultivated to obtain natural textile fibers and produce edible oil [1]. In arid and semi-arid zones around the world, high temperatures combined with drought and soil salinity influence the growth and productivity of cotton by affecting its morphological, physiological, and biochemical attributes [2]. Cotton is grown as a perennial shrub, so for better cotton production, high soil moisture content is essential because a lack of soil moisture content combined with soil salinity in the presence of high temperatures affects the plants’ photosynthetic rate, which will decrease their carbon uptake, which, in turn, eventually lowers the boll numbers and weight of the cotton plant and will eventually affect global cotton productivity [3]. It is expected that cotton yield will further decrease in the coming years in light of the continuous global climate change [4,5].

Abiotic stress negatively influences plant growth, yield, and productivity [6]. Soil salinity is the most damaging abiotic stress and severely affects agricultural productivity and worldwide food security [7]. Annually, the amount of cultivable land declines by 1–2% as a result of soil salinity. Globally, approximately 800 million hectares (23% of the total arable land) are impacted by soil salinity. By the year 2050, it is anticipated that half of the world’s arable land will be affected by salinity [8]. Salt-affected soil triggers numerous biochemical changes, such as increases in reactive oxygen species (ROS), the modulation of phytohormones [9], disturbances in ion uptake, and a reduction in photosynthetic processes [10]. Moreover, microbial activities, soil physicochemical properties, and enzymatic activities are constrained by salinity stress [11]. In addition, soil salinity causes a reduction in soil health, including decreased water conductivity and low organic matter [12]. Such dilemmas cause osmotic stress in cells, physiological drought, ionic toxicity, and nutrient insufficiency, leading to negative influences on metabolic function and decreased crop productivity [12,13]. So, it has become necessary to continuously search for new technologies to enhance crop growth and mitigate the hazardous impacts of soil salinity.

Freshwater insufficiency has proven to be a worldwide problem for sustainable agricultural development, particularly in arid and semi-arid zones [14]. Industrialization and urbanization can reduce the amount of available fresh water, intensifying the dangers of water scarcity stress on global agricultural production [14]. When cotton plants undergo water stress during the flowering and boll development stage, their yield may decrease by as much as 50% [15]. In addition, a limited soil moisture content reduces nutrient uptake, and reduced growth properties can lower the quality of cotton fibers [16]. Increasing the irrigation intervals during cotton growth causes a reduction in leaf water potential, which affects leaf enlargement [17]. Water stress negatively affects the growth of a cotton plant, its stomatal conductance, its leaf temperature, its carbon dioxide assimilation rate, and its chlorophyll content, which eventually decrease the amount of boll formation, the cotton yield, and fiber quality [16]. Therefore, using new techniques and implementing ecofriendly approaches are essential for sustaining future cotton plants tolerant to water-deficit stress [18].

In recent decades, studies on the application of bio-organic amendments have appealed for the introduction of alternative ecological technologies to mitigate salt stress and the deleterious impacts of water stress, thus improving crop productivity, such as the application of vermicompost to guarantee higher yields and fiber quality of cotton [19]. Vermicompost is a product of the non-thermophilic bio-degradation of organic materials through interactions between earthworms and microorganisms [20]. This organic compost is light, odorless, and free of weed seeds [21], offering abundant pores; high capacity for ventilation, drainage, and water storage [21]; large numbers of macro and micro elements [22]; great and active microbial biodiversity [22]; and many plant-hormone-like substances and growth regulators [23]. Consistent with these characteristics, vermicompost can play an effective role in plant growth and development and in decreasing the injurious impacts of different abiotic stresses [24]. Recently, new reports have stated that vermicomposting applications could enhance soil health by improving soil physicochemical properties [21], increasing the concentrations of plant-available nutrients [22], stimulating soil biological activities [24], and enhancing crop yield and/or quality due to enhancements in soil organic matter content that increase soil enzyme activity and due to the release of essential nutrients that counteract toxic ions in the soil solution [19].

Cyanobacteria are a diverse group of photosynthetic microorganisms that play an essential role in many ecosystems [25]. In recent years, there has been growing interest in their potential use as biofertilizers due to their ability to fix atmospheric nitrogen and produce various biostimulants, such as phytohormones and other growth-promoting substances [26]. Studies have reported positive effects of cyanobacteria on plant growth, such as increased shoot and root growth, chlorophyll content, and nutrient uptake [27,28]. One of the most well-known cyanobacteria is *Arthrospira platensis*, formerly known as *Spirulina*, which is a filamentous blue-green alga that has been extensively studied with respect to its nutritional and medicinal properties. In recent years, several studies have investigated the biostimulant effects of *Arthrospira* on plant growth and productivity [29]. *Arthrospira* contains a variety of biostimulants, including phytohormones, amino acids, vitamins, minerals, and antioxidants, which have been shown to promote plant growth and development, enhancing K^+^ uptake and reestablishing ion homeostasis under salinity stress [27]. The application of *Arthrospira* to plants can enhance photosynthesis, increase chlorophyll content, and improve plant growth parameters such as plant height, stem diameter, and leaf area [30]. Due to its gelatinous structure, *Arthrospira* can increase the soil’s ability to retain water [26]. In addition, *Arthrospira* can also improve soil microbial communities by promoting the growth of beneficial microorganisms, which can enhance nutrient cycling and improve soil fertility. Furthermore, *Arthrospira* contains antioxidants that protect plants against oxidative stress and improve their overall health [31]. Overall, the biostimulant properties of *Arthrospira* render it a promising natural alternative to chemical fertilizers and growth promoters. Further research is needed to explore its full potential in agriculture and optimize its application methods to maximize its benefits for plant growth and development.

The aim of the present trial was to assess the effects of an integrated application of vermicompost combined with cyanobacteria (*Arthrospira*) extract on soil traits, physiological properties, biochemical attributes, yield-related characteristics, and productivity alongside the nutritional value of barley plants irrigated with saline water and grown in salt-affected soil.

## 2. Results

### 2.1. Exchangeable Sodium Percentage (ESP) in the Soil and Ratio of K/Na in Cotton Leaves

Increasing the irrigation intervals to 21 and 28 days increased the amount of exchangeable Na in the soil by 25% and 57%, respectively (Figure 1A). However, the application of vermicompost, cyanobacteria, and their combined application under Irrig.14, Irrig.21, and Irrig.28 led to reductions in exchangeable Na percentages ranging from 12% to 46%, with the highest reduction achieved via the combined application of vermicompost and cyanobacteria under Irrig.21. These findings suggest that the application of vermicompost and cyanobacteria extract can improve soil properties and reduce soil salinity with regard to the growth of Egyptian cotton under salt-affected conditions.

The results revealed that increasing the irrigation intervals to 21 and 28 days reduced the potassium (K^+^) content in the plants’ leaves by 24% and 41%, respectively. However, the application of vermicompost and cyanobacteria extract, either alone or in combination, increased the K content. The combined application of vermicompost and cyanobacteria extract was found to be the most effective with respect to increasing the K content under different irrigation intervals. The combined application increased the K content over the control (Irrig.21) by 51%, whereas the percentage of increase was 15% when compared with the control treatment, Irrig.14 (Figure 1B). Moreover, increasing the irrigation intervals to 21 days and 28 days led to increases in the sodium (Na^+^) content in the cotton leaves by 36.6% and 66.0%, respectively (Figure 1C). However, the addition of vermicompost and cyanobacteria extract, either alone or in combination, reduced the Na content. The application of vermicompost reduced the Na^+^ content by 20%, 12%, and 9%, while cyanobacteria reduced the Na^+^ content by 26%, 17%, and 12%, respectively. The combined application of vermicompost and cyanobacteria extract was found to be the most effective with respect to reducing the Na content under different irrigation intervals, which reduced by 34%, 23%, and 18%, respectively (Figure 1C). The K/Na ratio was also decreased by increasing the irrigation intervals, but the combined application of vermicompost and the cyanobacteria extract increased the ratio (Figure 1D). Overall, the study showed that applying vermicompost and the cyanobacteria extract could improve soil quality and the nutrient content in plant leaves under salt-affected soil conditions with different water irrigation intervals.

### 2.2. Physiological Traits

#### 2.2.1. Photosynthetic Pigments

The drought effect, which resulted from increasing the irrigation intervals from 14 to 28 days, reduced the content of chlorophyll *a* and *b* in cotton leaves, which is indicative of a decline in a plant’s photosynthetic efficiency (Figure 2A,B). However, the application of vermicompost and the cyanobacteria extract led to an increase in the content of chlorophyll *a* and *b*, with the combined application of both resulting in the highest increase in chlorophyll content. The increase in chlorophyll content was more significant under moderate-deficit irrigation (every 21 days compared to its control) than the other two irrigation intervals compared to the control, Irrig.14, and Irrig.28, respectively. Moreover, the combined application of vermicompost and the cyanobacteria extract mitigated the adverse effects of osmotic stress on the chlorophyll *a* content under moderate-deficit irrigation but not under severe-deficit irrigation (every 28 days). The efficiency of chlorophyll *b* as an accessory photosynthetic pigment under different treatments and irrigation intervals was revealed via the ratio between chlorophyll *a* and *b* (Figure 2C). Increasing the irrigation intervals to 21 and 28 days resulted in a significant decrease in the Chl *b*/*a* ratio, with reductions of 17% and 48%, respectively. However, the application of vermicompost, cyanobacteria extract, and their combination significantly increased the Chl *b*/*a* ratios under all irrigation intervals. The combined application of vermicompost and the cyanobacteria extract under irrigation interval 21 resulted in the highest increase in the Chl *b*/*a* ratio, namely, 26%. Additionally, under severe-deficit irrigation (irrigation every 28 days), the combined application was incapable of compensating for the osmotic stress effect on the level of normal irrigation, resulting in an 18% reduction in the Chl *b*/*a* ratio compared to the control irrigation interval every 14 days (Figure 2C). Similarly, the carotenoid content showed a similar trend as that presented in the results regarding chlorophyll content (Figure 2D). Increasing the irrigation intervals to 21 and 28 days reduced carotenoid levels significantly. However, the soil application of vermicompost and the foliar application of the cyanobacteria extract, alone or in combination, increased carotenoid levels compared to the control under all irrigation intervals. The combined application of vermicompost and cyanobacteria extract under Irrig.21 showed the highest increase in carotenoid levels. However, under Irrig.28, the combined application was not able to compensate for the effects of osmotic stress, leading to a reduction in carotenoid levels. These findings suggest that the combined application of vermicompost and cyanobacteria can enhance the photosynthetic efficiency of Egyptian cotton plants under salt-affected soil, especially under moderate-deficit irrigation.

#### 2.2.2. Water Relations

Subjecting cotton plants to a gradual increase in irrigation intervals from 14 to 21 and 28 days resulted in reductions in relative water content (RWC) of 12% and 30%, respectively (Figure 3A). The application of vermicompost under Irrig.14 and Irrig.21 led to increases in RWC of 7% and 9%, respectively, while the application of cyanobacteria extract resulted in a slight increase in RWC under Irrig.14 and increases of 8% and 12% under Irrig.21 and Irrig.28, respectively. The combined application of vermicompost and the cyanobacteria extract resulted in a higher increase in RWC, with increases of 13%, 15%, and 18% under Irrig.14, Irrig.21, and Irrig.28, respectively. However, under severe water deficit conditions (Irrig.28), the combined application of vermicompost and cyanobacteria did not fully compensate for the effect of osmotic stress on the RWC, resulting in a reduction of 18% compared to the control treatment under Irrig.14 (Figure 3A).

The osmotic stress resulting from increasing the irrigation intervals to 21 days and 28 days significantly reduced stomatal conductance (gs) by 20% and 33%, respectively (Figure 3B). Vermicompost application and foliar spraying with the cyanobacteria extract alone or in combination showed a positive effect on gs. Vermicompost application under Irrig.14 presented an increase in gs by 6%, while foliar spraying with the cyanobacteria extract under Irrig.14 presented an increase in gs of 8%, and the combined application of vermicompost and cyanobacteria under Irrig.14 increased gs by 13%. Similar trends were observed for Irrig.21 and Irrig.28. Under Irrig.21, the application of vermicompost, the cyanobacteria extract, and their combined application increased gs by 11%, 16%, and 23%, respectively. Under severe water deficit conditions (Irrig.28), the application of vermicompost, cyanobacteria extract, and their combined treatment increased gs by 7%, 10%, and 15%, respectively. However, compared with the control, Irrig.14, the combined application under Irrig.21 recorded almost the same value, whereas the combined application under Irrig.28 showed a reduction in gs by 23% (Figure 3B). These findings suggest that the combined application of vermicompost and cyanobacteria extract could improve plant water status under moderate water deficit conditions but may not be sufficient to alleviate severe water stress.

#### 2.2.3. Oxidative Stress Indicators

As the irrigation intervals increased to 21 days and 28 days, the cotton plants showed significant increases in the level of H_2_O_2_ by 24% and 47%, respectively (Figure 4A). However, with the application of vermicompost and the cyanobacteria extract, whether separately or in combination, the level of H_2_O_2_ in the plants decreased substantially. Under the recommended irrigation interval of every 14 days, the application of vermicompost and cyanobacteria extract resulted in a reduction in the concentration of H_2_O_2_ by 34% and 41%, respectively, while the combined use of both resulted in a remarkable reduction of 61%. Similarly, under moderate-deficit irrigation every 21 days, using vermicompost, the cyanobacteria extract, and their combination reduced the H_2_O_2_ concentration by 24%, 34%, and 57%, respectively. Even under severe-deficit irrigation every 28 days, vermicompost and the cyanobacteria extract still managed to reduce the H_2_O_2_ concentration by 18% and 22%, respectively, while the combined application resulted in a reduction of 29%. However, it was observed that the combined application could not fully compensate for the effect of osmotic stress caused by severe water deficit and increased the H_2_O_2_ concentration by 4% (Figure 4A).

The level of malondialdehyde (MDA), a marker of oxidative stress in plants, increased under irrigation every 21 and 28 days by 89% and 156%, respectively (Figure 4B). However, the application of vermicompost and the cyanobacteria extract, either alone or in combination, reduced the level of MDA in the plants. Under the recommended irrigation interval of every 14 days, the application of vermicompost and the cyanobacteria extract reduced the level of MDA by 34% and 53%, respectively, while the combined application resulted in a reduction of 63%. Similarly, under moderate-deficit irrigation (21 days), vermicompost, cyanobacteria, and their combined application reduced MDA levels by 25%, 33%, and 55%, respectively. Under severe-deficit irrigation (28 days), vermicompost and the cyanobacteria extract also reduced MDA levels by 15% and 22%, respectively, while their combined application reduced the MDA concentration by 39%. Compared to the control treatment under Irrig.14, the combined application under Irrig.21 was able to reduce MDA levels by 16%. However, the combined application under Irrig.28 was not able to compensate for the osmotic stress effect on the level of normal irrigation, for which an increase in MDA levels by 55% was recorded. These findings underscore the importance of the application of vermicompost and a cyanobacteria extract, especially in combination, in mitigating the harmful effects of oxidative stress on cotton plants grown in salt-affected soils with limited water availability.

#### 2.2.4. Antioxidant Defense System

##### Non-Enzymatic Antioxidants

The increasing irrigation intervals of the drought-stressed cotton plants led to increased levels of amino acids and proline, with the highest increases observed at 21 and 28 days (Figure 5A,B). Under irrigation every 14 days, the vermicompost treatment, cyanobacteria extract treatment, and their combined application increased the concentration of amino acids by 17%, 226%, and 34%, respectively. Under irrigation every 21 days, the vermicompost treatment increased the content of amino acids by 9%, while the cyanobacteria extract treatment increased this value by 16%. The combined application of vermicompost and the cyanobacteria extract under irrigation every 21 days increased the concentration of amino acids by 21%. Under irrigation every 28 days, the vermicompost treatment and cyanobacteria extract treatment increased amino acid levels by 15% and 21%, respectively. The combined application of vermicompost and the cyanobacteria extract under irrigation every 28 days increased amino acid levels by 28%. The effects of the different treatments on the plants’ free amino acid content were more pronounced under moderate drought stress (Irrig.21) than under severe drought (Irrig.28). Under irrigation every 21 and 28 days, the combined application effected the highest increase in proline levels, namely, by 126% and 163%, respectively, compared to the control of irrigation every 14 days (Figure 5A,B).

After increasing the irrigation intervals to 21 days (15% reduction) and 28 days (34% reduction), a marked reduction in the concentration of total soluble sugars (TSS) in the cotton leaves was recorded (Figure 5C). However, the application of vermicompost, cyanobacteria, and a combination of both under Irrig.14 resulted in a slight increase in TSS by 6%, 11%, and 19%, respectively. These organic supplements proved to be effective in increasing TSS levels under Irrig.21, with vermicompost, cyanobacteria, and a combination of both presenting increases of 12%, 15%, and 20%, respectively. Under Irrig.28, these supplements were even more effective, with vermicompost, cyanobacteria, and a combination of both effecting increases in TSS levels of 14%, 16%, and 32%, respectively. When compared to the control, Irrig.14, the application of the combined treatment under Irrig.21 slightly increased TSS levels by 1.6. However, the combined treatment under Irrig.28 was not able to compensate for the effect of osmotic stress on the level of normal irrigation, resulting in a 13% reduction in the level of TSS (Figure 5C).

Significant increases in total phenolics were observed with both extended irrigation intervals and the application of organic supplements (Figure 5D). Irrigation every 21 days increased total phenolics by 17%, but the real breakthrough was observed with regard to 28-day intervals, which led to a significant increase in total phenolics (38%). Vermicompost and cyanobacteria under Irrig.14 each contributed to smaller but still noteworthy increases in total phenolics, with their combined application resulting in a more substantial increase of 20%. The results were even more impressive under Irrig.21, where vermicompost and cyanobacteria each led to further increases in total phenolics, but their combined application presented a staggering jump to 26%. Compared to the control, Irrig.14, the combined application under Irrig.21 produced an impressive 47% increase in total phenolics, while under Irrig.28, the same combined application resulted in a remarkable increase of 69% (Figure 5D).

##### Enzymatic Antioxidants

Significant increases in superoxide dismutase (SOD) activity of 18% and 32% in the leaves of the cotton plants were recorded after increasing the irrigation intervals to 21 days and 28 days, respectively (Figure 6A). Furthermore, under regular irrigation (Irrig.14), the application of vermicompost and the cyanobacteria extract increased SOD activity by 9% and 19%, respectively, while their combined application increased it by 24%. Similar trends were observed under moderate-deficit irrigation (Irrig.21), with increases of 6%, 17%, and 19% for vermicompost, the cyanobacteria extract, and their combined application, respectively. Under severe-deficit irrigation (Irrig.28), the vermicompost and the cyanobacteria extract applications resulted in increases of 11% and 15%, respectively, while their combined application resulted in an increase of 22% in SOD activity. When compared to the non-treated control under regular irrigation, the combined application under moderate-deficit irrigation raised the activity level of SOD to 40%, whereas under severe-deficit irrigation, the combined application resulted in the highest increase in SOD activity, amounting to 61%.

Increasing the irrigation intervals to 21 and 28 days resulted in significant increases in peroxidase (POD) activity of 46% and 128%, respectively (Figure 6B). The use of vermicompost and the cyanobacteria extract, either individually or in combination, also led to increases in POD activity, wherein the vermicompost treatment showed a better effect than cyanobacteria under all irrigation intervals. The combined application of vermicompost and the cyanobacteria extract showed the highest increases in POD activity compared to the control treatment, increasing POD activity by 2.5-fold under Irrig.21 and 3.5-fold under Irrig.28 (Figure 6B).

The activity of catalase in the leaves of the cotton plants was reduced by 22% and 56% as a result of prolonged irrigation for 21 and 28 days, respectively (Figure 6C). However, the addition of vermicompost under regular irrigation (Irrig.14) resulted in a slight increase of 8%, while foliar spraying with cyanobacteria extract showed a 15% increase, and the combined application increased catalase activity by 36% under the same irrigation interval. Similar results were observed under moderate-deficit irrigation (Irrig.21), where vermicompost, the cyanobacteria extract, and their combination increased catalase activity by 17%, 29%, and 38%, respectively. Under severe-deficit irrigation (Irrig.28), the combined application of vermicompost and the cyanobacteria extract was not able to compensate for the effect of osmotic stress on the level of normal irrigation, resulting in a 29% reduction in catalase activity compared to the control, Irrig.14 (Figure 6C).

### 2.3. Vegetative Characteristics

Subjecting the cotton plants to moderate or severe osmotic stress reduced the plants’ height by 13% and 19% and their total leaf area by 19% and 36%, respectively (Figure 7). Slight increases in plant height were observed under irrigation every 14 days for all biostimulant treatments (vermicompost, cyanobacteria, and combination). Under normal irrigation conditions, vermicompost and cyanobacteria induced increases in total leaf area of 24% and 39%, respectively, while their combined application effected the highest increase (45%). Vermicompost, cyanobacteria, and their combined application under Irrig.21 increased plant height by 12%, 9%, and 15%, respectively, and total leaf area by 21%, 33%, and 40%, respectively. Under irrigation every 28 days, the combined application was not able to compensate for the effect of osmotic stress on the level of normal irrigation and precipitated a slight reduction in plant height (4%) and total leaf area (13%). The combined application under irrigation every 21 days resulted in a significant increase in leaf area (14%) compared to the control, Irrig.14 (Figure 7).

### 2.4. Yield Characteristics

Increasing the irrigation intervals for the cotton plants resulted in a significant decrease in the number of fruiting branches observed, presenting a 42% reduction under severe-deficit irrigation (Table 1). However, the application of vermicompost, the cyanobacteria extract, and their combination significantly increased the number of fruiting branches under all irrigation intervals. The combined application showed the highest increase in fruiting branches under all irrigation intervals, except for severe-deficit irrigation, wherein it could not compensate for the effect of osmotic stress on the level of regular irrigation.

Significant reductions in the number of open bolls per plant and boll weight were noted with moderate and severe-deficit irrigation (Irrig.21 and Irrig.28). However, the combined application of vermicompost and the cyanobacteria extract mitigated the negative effect of drought conditions on the number of open bolls and boll weight. Under the irrigation interval of 21 days, the combined application resulted in the highest increase in the open boll number per plant (45%) and boll weight (9%) compared to the other treatments. On the other hand, under severe-deficit irrigation (28 days), the combined application was not able to compensate for the effect of osmotic stress on the level of regular irrigation, which resulted in a 23% reduction in the number of open bolls per plant and a 5% reduction in boll weight (Table 1).

The results in Table 1 show that under irrigation every 21 and 28 days, the lint% reduced by 10% and 16%, respectively. Under the irrigation interval of 14 days, vermicompost and the cyanobacteria extract slightly increased the lint percentage by 3% and 2%, respectively, while the combined application increased it by 6%. Under the irrigation interval of 21 days, vermicompost, the cyanobacteria extract, and their combined application increased the lint percentage by 9%, 7%, and 11%, respectively. Under the irrigation interval of 28 days, the combined application was not able to compensate for osmotic stress, resulting in a slight reduction in lint percentage by 5% compared to the control irrigation interval of 14 days.

Increasing irrigation intervals from 14 to 21 and 28 days reduced seed yield by 13% and 24%, respectively (Table 1). However, the application of vermicompost and cyanobacteria extract, alone or combined, improved seed yield under all irrigation intervals. Specifically, under irrigation every 14 days, vermicompost increased seed yield by 13%, cyanobacteria extract increased it by 10%, and the combined application resulted in an increase of 19%. Under irrigation every 21 days, vermicompost increased seed yield by 10%, cyanobacteria extract increased it by 8%, and the combined application resulted in an increase of 18%. Under irrigation every 28 days, the combined application was unable to compensate for osmotic stress, resulting in a reduction in seed yield by 10% compared to the control under irrigation every 14 days, whereas the combined application under Irrig.21 resulted in a 2% increase in seed yield compared to the control under irrigation every 14 days (Table 1). These findings highlight the potential of using vermicompost and cyanobacteria extract to improve cotton plant growth and productivity under different water irrigation intervals in salt-affected soil.

The fiber quality of the Egyptian cotton plants (cv. Giza 94) grown in salt-affected soil was significantly affected by the stress conditions induced by the non-recommended irrigation intervals (Figure 8). Increasing the irrigation intervals to 21 days and 28 days significantly reduced the Pressley index (by 13% and 31%) and Micronaire reading (by 22% and 43%) of the cotton plants, respectively. The application of vermicompost and the cyanobacteria extract significantly increased the Pressley index and Micronaire reading under different irrigation intervals, especially when applied together under moderate-deficit irrigation every 21 days. The combined application increased the Pressley index by 19% and the Micronaire reading by 37% compared to the non-treated control under the same irrigation interval. Although the combined application under severe-deficit irrigation every 28 days generated increases in both the Pressley index by 29% and the Micronaire reading by 46% compared to the non-treated control, it failed to restore the fiber quality to its normal value under normal irrigation, which resulted in a reduction in the Pressley index by 11% and the Micronaire reading by 17%. Overall, the study suggests that the combined application of vermicompost and cyanobacteria extract could be a potential approach to improving the fiber quality and strength of cotton plants under moderate water deficit conditions.

## 3. Discussion

### 3.1. Effect of Soil and Foliar Treatments on Soil Exchangeable Sodium Percentage (ESP) under Osmotic Stress

Numerous investigations have been carried out to enhance the growth and yield of cotton under abiotic stresses (i.e., soil salinity and water stress), including the application of plant growth regulators and micronutrients in addition to foliar and soil applications, but this aim remains a significant challenge, specifically in arid areas. However, the integration of vermicompost and cyanobacteria extract has received less attention thus far, thereby confirming the importance of the present examination, particularly with respect to the effect of increasing irrigation water intervals, in salt-affected soil, on the growth and productivity of cotton plants.

Under both saline soil and water stress conditions, the increased exchangeable sodium percentage (ESP) resulted in a reduction in the plants’ growth, which might have been due to osmotic injury or specific ion toxicity caused by the uptake of salt as well as a decline in water and nutrient uptake, thus reducing soil quality, plant growth, and crop productivity [32]. Our results showed that the integrated application of vermicompost and *Arthrospira* extract significantly enhanced cotton growth, thereby mitigating salt and water stress by decreasing ESP and increasing then quantity of Ca, Mg, and K ions, which positively influence water and nutrient uptake [33].

In addition, it was found that vermicompost has a positive effect on cotton growth and development under increasing irrigation intervals and in salt-affected soil. Vermicompost has been deemed a carbon source for microbes that may enhance soil organic carbon content, leading to an improvement in the physical and chemical attributes of soil and a decrease in oxidative stress [34], thereby enhancing cell division and expansion and enabling a plant to absorb growth-regulatory nutrients required for processes such as photosynthesis, protein synthesis, and lipid metabolism and that eventually positively affect plant height [34]. Through this current study, it is clear that the employed cyanobacteria extract was effectively integrated as a natural product, for which its content of organic matter, macro- and microelements, and some plant growth regulators proved to have beneficial effects on stimulating leaf area and plant height under increasing irrigation intervals in salt-affected soil [27]. It was observed that the integrated application of vermicompost and the cyanobacteria extract had a higher synergistic impact on decreased ESP and increased leaf area and plant height under water stress and soil salinity than their individual application.

### 3.2. Effect of Soil and Foliar Treatments on Physiological Attributes under Osmotic Stress

It was demonstrated that the cotton plants exposed to increasing irrigation intervals presented reduced photosynthetic rates as a result of increased concentrations of Na^+^ and Cl^−^ ions in the leaves, stomatal closure, and decreased chlorophyll content and relative water content, resulting in dry weight loss, increased osmotic stress, and ion toxicity [35]. However, it was observed that the application of vermicompost could increase the content of chlorophyll *a* and *b*, carotenoids, and K ions and decrease the content of Na ions in the cotton leaves while increasing relative water content and stomatal conductance. These findings are consistent with those uncovered by Cevheri et al. [36]. All these benefits occurred as a result of the stimulation of auxin-like substances produced during vermicompost consumption. Vermicompost composition involves humic and fulvic acids and other organic acids along with nutrients, particularly nitrogen, that can promote plant growth and development [24]. The humic substances in vermicompost have a high absorption capacity for nutrients as a result of the presence of negatively charged functional groups [37]. It has also been reported that auxin and other plant growth hormones have been found in vermicompost [38]. Generally, it appears that the physical, chemical, and biological structure of vermicompost has provided better conditions for water and nutrient uptake [39]. Vermicompost can augment the amount of water entering roots due to its capacity to hold water via augmented root hydraulic conductivity [40]. In this respect, some reports have indicated that plant hormones and calcium play crucial roles in regulating stomata [40]. These findings prove that vermicompost can efficiently enhance water potential of cotton leaves owing to its content of plant hormones and organic ions, porous structure, and high water-holding capacity, which increase the absorption of K^+^. Reports have stated that the cytokinin hormone can augment K^+^ uptake [41], and vermicompost includes plant growth hormones such as cytokinin [38]. So, vermicompost can enhance nutrient uptake, especially K^+^, and decrease Na^+^ uptake under combined drought and salt stress, resulting in the mitigation of the adverse impacts of abiotic stress [41].

Additionally, cyanobacteria extracts are attaining appeal as cost-benefit substitutes to conventional fertilizers due to their environmentally friendly and non-toxic characteristics [26]. The K^+^ content in the extract of cyanobacteria is high, which is necessary for sustaining the balance of photosynthesis under salinity and water stress [30]. Cyanobacteria extract has been reported to encourage the buildup of nonstructural carbohydrates, which enhance energy storage while increasing metabolism, and enhance leaf growth by improving leaf water retention, via the transport of osmolytes/ions, which increases tolerance to water stress in salt-affected soil [29].

### 3.3. Effect of Soil and Foliar Treatments on Antioxidant Enzyme Activity and Oxidative Indicators under Osmotic Stress

A chemical composition analysis of the cyanobacteria extract isolated from *Spirulina* revealed that the extract is rich in nutrients due to its content of antioxidants (CAT, SOD, and POD) [42]. These antioxidants combat abiotic stress and play a crucial role in inhibiting toxic free radicals (H_2_O_2_ and MDA) from accumulating while enhancing crop growth and productivity by augmenting nutrient absorption [42]. It was found that foliar spraying with cyanobacteria extract can mitigate the reduction in plant growth caused by sodium chloride as a result of a direct increase in cell division and expansion. This beneficial effect of cyanobacteria extract can be attributed to the fact that it contains growth-promoting substance and high concentrations of total carbohydrates, proline, and phenolic compounds that induce higher crop yields [31].

The improved activity of the antioxidant enzymes upon the combined application of vermicompost and the cyanobacteria extract facilitated the conversion of H_2_O_2_ into non-toxic compounds (H_2_O and O_2_), thereby protecting the plants from the harmful impacts of water stress on cell membranes and macromolecules under salt-affected soil and decreasing MDA levels compared to the individual treatments [43,44]. The cotton plants exposed to water stress in salt-affected soil and treated with the cyanobacteria extract presented boosted vegetative growth attributes and enhanced photosynthesis processes that were positively associated with the photosynthetic pigment (chlorophyll *a* and *b* as well as carotenoids) concentrations in the leaves.

It was found that cyanobacteria extract from *Spirulina* contained high levels of Mg, Fe, and N (structural components of chlorophyll), which may augment chlorophyll biosynthesis under abiotic stress. Additionally, a high content of total nonstructural carbohydrates (linked to increased photochemical efficiency and proline) was noted, whose presence resulted in higher photosynthetic activity [45,46].

The cotton plants exposed to water stress in salt-affected soil and treated with the cyanobacteria extract presented increased translocation rates of various photosynthetically produced molecules to fruits, constituting a process that was positively linked with higher yields. Related findings have been described for various plants [30,47,48]. Applying the cyanobacteria extract induced a decrease in the acquisition of chlorine ions in the leaves of cotton plants exposed to water stress in salinity-affected soils, which probably led to an increase in the relative water content. The sodium ion content in plant leaves can decrease, while the cytosolic potassium ion content increases by releasing a K^+^ vacuolar [49], which is osmotically counterbalanced by K^+^ in the cytosol, and this may be a necessary mechanism by which cotton plants tolerate salinity stress.

### 3.4. Effect of Soil and Foliar Treatments on Osmolytes under Osmotic Stress

Cyanobacteria extract was used to reduce the anticipated degradation due to abiotic stress, which is mainly due to oxidative damage. Additionally, phenolic and flavonoid content, antioxidant capacity, and proline accumulation were observed to increase [25]. This is attributed to the growth-stimulating activities and composition of the cyanobacteria extract itself. Cyanobacteria extract contains macroelements, microelements, vitamins, auxins, and phytohormones, which have antioxidant properties. The bioactive compounds of the cyanobacteria extract from *Spirulina platensis*, which contain phenols, carbohydrates, protein, and proline, positively affect overall plant growth by improving metabolite production, which improves tolerance to abiotic stress via the modulation of water capacity and turgor pressure in plants [26]. Proline (an amino acid) plays a highly beneficial role in osmoregulation as an indicator of stress; additionally, it acts as an excellent osmolyte and has been reported to exhibit antioxidant activity under abiotic stresses such as water stress and soil salinity, which increase its concentration noticeably under abiotic stress [50,51]. It was observed in our study that the use of cyanobacteria extract induced higher proline content and decreased the levels of lipid peroxidation under stress conditions compared to the untreated plots. This result shows the effective role of the cyanobacteria extract as a scavenger of ROS, contributing to cellular homeostasis and decreasing membrane damage, which mitigates the adverse effects of water stress in salinity-affected soils [52]. In addition, it was observed that the use of the cyanobacteria extract on cotton plants improved the activity of antioxidant enzymes such as CAT, SOD, and POD under water stress conditions in salt-affected soil. The findings indicated that the use of the cyanobacteria extract resulted in enhanced photosynthetic pigment levels (carotenoids and Chl *a* and *b*), phenolic content, proline content, and metabolite levels (e.g., total soluble sugars, proline, and total free amino acids), which provide the energy needed to activate defense strategies that are known ROS non-enzymatic scavengers and whose activity was increased under water stress conditions in salt-affected soil [53]. In the current investigation, the Spirulina-treated cotton plants showed a significant improvement in CAT, SOD, and POD activity compared to that of the untreated plants, which eventually improved growth and yield under abiotic stress [31]. Cyanobacteria extract includes compounds rich in antioxidants, cytokinins, auxin-like growth substances, carbohydrates, polyphenols, proline, and amino acids, which combat abiotic stress and play a beneficial role in inhibiting ROS, thus enhancing the growth and productivity of cotton by augmenting nutrient uptake [26] while improving its tolerance to abiotic stress.

### 3.5. Effect of Soil and Foliar Treatments on Yield-Related Traits and Productivity under Osmotic Stress

The current investigation reveals the important impacts of the cyanobacteria extract from *Spirulina*, whose influence significantly augments the number of fruiting branches/plant, the number of total flowers/plants, the number of open bolls per plant, and boll weight. These impacts have a positive effect on improving seed cotton yield/hectare as well as fiber quality (Pressley index and Micronaire reading) as observed under stressful environmental conditions. These yield data correspond with those obtained Yanni et al. [54], thereby highlighting a strong positive correlation between cyanobacteria extract and the augmentation of the number of fruiting branches/plant, the number of total flowers/plants, the number of open bolls/plant, and boll weight. This yield increment is believed to be due to phytohormones that exist in the cyanobacteria extract containing cytokinins and the induction of host hormonal synthesis [28]. Furthermore, our findings confirmed that the extract of *Spirulina* can be used as a promising eco-friendly and multifunctional agricultural practice for improving sustainable agricultural production. As shown in the current investigation, the integrated application of vermicompost as a soil application alongside cyanobacteria extract through foliar spraying is a promising technique with which to further increase the growth and yield characteristics of cotton plants and mitigate the harmful impacts of abiotic stress.

## 4. Materials and Methods

### 4.1. Experimental Layout and Treatments

During two successive summer growing seasons in 2020 and 2021, a field experiment was carried out in Elamaar village in the region of Sidi Salem (31°07 N, 30°57 E), Kafr El-sheik Governorate, Egypt, to assess the effects of treating soil with vermicompost and executing foliar spraying with cyanobacteria extract (*Arthrospira platensis*) on the soil properties, growth, physiological traits, biochemical attributes, antioxidant enzyme activity, productivity, and the fiber quality of Egyptian cotton plants (*Gossypium barbadense* L. cv. Giza 94) under different water irrigation intervals in salt-affected soil. The three water irrigation intervals tested are as follows: the recommended irrigation interval was every 14 days (Irrig.1), which was termed regular irrigation herein; moderate-deficit irrigation was every 21 days (Irrig.2); and severe-deficit irrigation was every 28 days (Irrig.3). Each irrigation treatment was stopped one month before harvesting. Additionally, four amendment treatments were employed, including a non-treated control, in which the vermicompost or cyanobacteria extract were not applied; soil application of vermicompost; foliar application of cyanobacteria extract; and combined application of vermicompost and cyanobacteria extract. The structure of the experimental assay corresponded to a randomized complete block design, with treatments arranged in split plots with four replications. Treatments of irrigation water (every 14, 21 and 28 days) were placed in the main plots, while amendment treatments (control, vermicompost, cyanobacteria extract, and vermicompost + cyanobacteria) were placed in the sub-plots. Seeds (70 kg ha^−1^) were obtained from Cotton Research Institute, Agriculture Research Center, Egypt. Each plot consisted of 5 rows, which were 4 m long and 70 cm apart and 30 cm among hills. Sowing took place on 8th April, leaving two plants/hill at thinning time in both seasons. Typical culturing practices were employed throughout the growing seasons. Phosphorus fertilizer was applied at a rate of 53.5 kg P_2_O_5_ ha^−1^ as calcium superphosphate (15.5% P_2_O_5_) during land preparation. Nitrogen fertilizer, at a rate of 107 kg N ha^−1^ as ammonium nitrate (33.5% N), was added in two equal doses immediately before the first and the second irrigations. Potassium fertilizer in the form of potassium sulphate (48% K_2_O) was added as soil application at the rate of 120 kg ha^−1^. Temperature, wind speed, and relative humidity data throughout both seasons are presented in Table 2.

Some physical and chemical analyses of the soil at the experimental farm are given in Table 3.

#### 4.1.1. Vermicompost Characterization (VC)

Vermicompost was prepared in a vermicomposting bin with dimension of 100 × 120 × 50 cm. Crop residues (rice and maize straw) were used as materials for VC. Cow manure and green waste were used as worm feed. Two earthworm species (*Eisenia fetida* and *Dendrobaena veneta*) were also inoculated [55]. Vermicompost was obtained from the Central Lab. of the Agric. Climate, Agric. Res. Centre, Giza, Egypt. The characteristics of the vermicomposting process are as follows: moisture content was maintained at 80% (volume) throughout the vermicomposting process (two months). The specifications of the chemical analysis of the prepared vermicompost are as follows: organic matter—42%; EC—3.8 dS m^−1^; total N—2.1%; total P—7.8%; total K—0.5%; pH—7.4; polyphenol content—8.4%; water-holding capacity—150 ± 12.23; and C content—18.3 ± 1.36%. The vermicompost was applied to the soil at a rate of 10.0 t ha^−1^.

#### 4.1.2. Cyanobacteria Extract Characterization

Fresh cyanobacteria extract (*Spirulina platensis*) was obtained from the Microbiology Department of the Soils, Water, and Environment Institute, Sakha Agricultural Research Station, Kafr EL-Sheikh Governorate, Egypt. Two grams of dry powder from the cyanobacteria of fresh cyanobacteria extract (*Spirulina platensis*) was added into one litre of deionized water, regularly mixed for fifteen minutes, and autoclaved at 121 °C for 60 min at 1.21 kg cm^−2^ [48]. The warm extract was purified using Whatman No. 40 filter paper from Sigma-Aldrich (Merck KGaA, Darmstadt, Germany) and stored at −4 °C until additional analysis. *Spirulina* extract contains eighteen amino acids (%): alanine 2.62; arginine 1.96; aspartic 3.4; cysteine 1.71; glutamic acid 3.82; glycine 1.82; histidine 0.45; isoleucine 1.59; leucine 2.55; lysine 1.35; methionine 1.05; phenylalanine 1.77; proline 1.17; serine 1.22; tryptophan 1.70; threonine 2.66; tyrosine 1.14; and valine 2.09. Their chemical compositions are as follows: oligosaccharide (3%); alginic acid (5%); phytin (0.003%); menthol (5%); natural growth regulators such as cytokines (0.001%), indole acetic acid (0.0002%), and pepsin (0.02%); and minerals (potassium oxide (18%), phosphorus oxide (5%), N (5%), Ca (1.5%), Zn (0.3%), Fe (2%), and Mn (0.1%)). The foliar solution consisted of 2.4 L from *Spirulina* extract and 250 L of water ha^−1^ at 30 and 50 days from sowing.

### 4.2. Exchangeable Sodium Percentage (ESP) in Soil

Soil samples were collected from various treatments of the experimental field at maturity by an auger and put in an oven for 24 h to estimate soil Na^+^, Ca^2+,^ and Mg^2+^ in soil solution (meq L^–1^) as saturated paste extract using Atomic Absorption Spectrophotometer (AAS, PERKIN ELMER 3300) to estimate soil sodium adsorption ratio (SAR). ESP was calculated as described by Arshad et al. [56]:ESP = 1.95 + 1.03 × SAR (R^2^ = 0.92)

### 4.3. Determination of Na^+^ and K^+^ in Cotton Leaves

At day 80, after the seed were sown, the second fully-expanded leaf of each treatment was sampled, dried, ground into fine powder, and digested using a combination of sulfuric acid and perchloric acid. The resulting solution was raised to a volume of 50 mL with ultra-pure water and analyzed using AAS to determine Na^+^ and K^+^ content as per the method described by Temminghoff and Houba [57].

### 4.4. Physiological Traits

#### 4.4.1. Chlorophylls and Carotenoids

The chlorophyll *a* and *b* content and carotenoid levels in the cotton leaves were determined according to the method described by Lichtenthaler and Wellburn [58]. To eliminate any impurities prior to extraction, a total of five newly picked leaves were cleaned. Then, 0.5 g from each sample was taken and ground with a mortar and pestle in 80% acetone. The resulting mixture was centrifuged, and the absorbance was measured at 663, 645, and 470 nm using a spectrophotometer. The amounts of chlorophyll *a*, chlorophyll *b*, and carotenoids were determined and reported as mg g^−1^ FW.

#### 4.4.2. Water Relations

##### Relative Water Content (RWC)

To determine the relative water content (RWC) in cotton leaves, leaf discs (6 mm in diameter) were obtained from 80-day-old plants. The discs’ fresh weight (FW) values were determined; then, they were immersed in distilled water at 25 °C for 24 h to obtain their turgid weight (TW). The discs’ dry weight (DW) was measured after drying them in a forced-air oven at 80 °C for 24 h. The RWC was calculated using the equation developed by Weatherley [59], as follows:$$RWC=\frac{FW-DW}{TW-DW}\times 100$$

##### Stomatal Conductance (gs; mmol H_2_O m^−2^ s^−1^)

At 80 days after cultivation, stomatal conductance was assessed with respect to the top-most fully expanded leaf using a dynamic diffusion porometer (Delta-T AP4, Delta-T Devices Ltd., Cambridge, UK) applied to the abaxial and adaxial surfaces of three leaves.

#### 4.4.3. Stress Indicators (Lipid Peroxidation and H_2_O_2_)

Malondialdehyde (MDA) levels were measured to determine lipid peroxidation in frozen cotton leaves according to the method described by Du and Bramlage [60]. The frozen leaf samples (0.5 g) were homogenized in 5 mL of 0.1% trichloroacetic acid (TCA) and centrifuged at 10,000× *g* for 15 min. The supernatant (1 mL) was mixed with 4.0 mL of 0.5% thiobarbituric acid (TBA) in 20% TCA solution, boiled for 30 min, and then quickly cooled in an ice bath. The absorbance of the supernatant was recorded spectrophotometrically at 532 and 600 nm. MDA content was calculated using the molar extinction coefficient of MDA ($1.56\times 10^{5}$ M^−1^ cm^−1^) and expressed as nmol g^−a^ FW.

To determine the concentration of hydrogen peroxide (µmol g^−1^ FW) in cotton leaves, the method reported by Velikova et al. [61] was used. Frozen leaf samples (0.5 g) were homogenized in 5.0 mL of 0.1% TCA, and the resulting supernatant was centrifuged at 10,000× *g* for 15 min. The H_2_O_2_ content was determined by mixing 0.5 mL of supernatant with 0.5 mL of 10 mM K-phosphate buffer (pH 7.0) and 1 mL of 1 M KI to initiate the reaction and then measuring the absorbance spectrophotometrically at 390 nm. The H_2_O_2_ content was calculated using the standard curve of H_2_O_2_.

#### 4.4.4. Antioxidant System

##### Non-Enzymatic Antioxidants

The total soluble sugars, free amino acids, total phenols, and proline content of cotton leaves were determined using various methods. The anthrone method described by Loewus [62] was used to determine the total soluble sugars in the ethanol extract of the leaves. For the determination of total free amino acids, the ninhydrin method described by Rosen [63] was employed, for which the ethanol extract of the leaves was used. The Folin–Ciocalteau method, as described by Singleton et al. [64], was used to determine the total phenols in the ethanol extract of the leaves. Lastly, the proline content of the cotton leaves was determined using the method reported by Bates et al. [65].

##### Enzymatic Antioxidants

At 80 days after cultivation, leaf discs (1 g) from the top-most fully expanded leaf were crushed and mixed in 5 mL of cold phosphate buffer (50 mM phosphate buffer at pH 7.0, comprising 0.5 mM EDTA and 1% polyvinylpolypirrolidone) for their exploitation as an enzyme extract, which was centrifuged for 25 min at 10,000× *g* at 4 °C [51,66].

Superoxide dismutase (SOD: 1.15.1.1) activity was assessed using nitro-blue tetrazolium (NBT) photochemical examination at 560 nm according to the process reported by Beauchamp and Fridovich [67].

The activity of catalase (EC 1.11.1.6) was determined using the method outlined by Aebi [68]. Prior to analysis, an enzyme extract was prepared using 0.5 mL of 0.2 M H_2_O_2_ in 10 mM K-phosphate buffer (pH 7.0). The activity of catalase was determined at 240 nm by using a spectrophotometer to measure the level of H_2_O_2_ consumed. The specific activity of catalase was expressed as Unit mg^−1^ protein.

The activity of peroxidase (POD: 1.11.1.7) was determined using o-phenylenediamine as a chromogenic indicator in the presence of H_2_O_2_ and enzyme extract according to the method described by Vetter et al. [69]. The activity was expressed as Unit mg^−1^ protein.

### 4.5. Vegetative Growth Attributes

#### 4.5.1. Plant Height

Plant height was recorded from the bottom of a plant at the soil surface to the highest point on the stem in all treatments at the stage of physiological maturity.

#### 4.5.2. Total Leaf Area

Ten samples per treatment were randomly collected and washed twice. Subsequently, 20 leaf discs (1 cm^2^ each) were dehydrated after having been placed in an oven at 85 °C for one day to determine the dry weight of the disks (DDW). The total leaf area/plant was computed using the following formula, which was developed by Wallace and Munger [70]:$$Totalleafareaperplant=\frac{LDW}{DDW}\times DA$$

LDW is the total leaf dry weight (g). 

### 4.6. Cotton Yield Attributes

#### 4.6.1. Growth and Yield Parameters

At the time of picking, ten plants were randomly chosen to estimate the number of fruiting branches plant^−l^, boll weight, and number of bolls plant^−1^. The obtained plants were dehydrated using air and, therefore, weighed separately. Twenty samples were obtained randomly and labelled in each experimental unit, and number of fruiting branches/plant and number of total flowers/plants were determined manually. Finally, means were computed for each replication. The bolls plant^−1^ were detached and their numbers were enumerated from the randomly chosen samples at harvest and averaged to obtain the number of bolls plant^−1^. For the average boll weight (g) data, 20 effective and opened bolls were obtained randomly from each treatment. The average boll weight was calculated and stated in grams (g). The seed cotton yield per hectare was computed from the net plot area and seed cotton weight of 20 detached bolls. Seed cotton yield of each plot was computed as kg ha^−1^.

#### 4.6.2. Fiber Properties

Micronaire reading to assess fitness and the Pressley index to assess strength were applied to the ten samples using the A.S.T.M. method [71].

### 4.7. Statistical Analysis

The two-year split-plot experiment data were statistically analyzed using CoStat software (Package 6.45, CoHort, Monterey, CA, USA) and analysis of variance (ANOVA). Tukey’s range test was used as a post hoc pairwise comparison to compare the means for any differences, with a significance level set at *p* < 0.05.

## 5. Conclusions

The application of foliar spraying with cyanobacteria-based bioproducts and the application of vermicompost to soil stimulated crop production systems as a consequence of their unique constituents and impacts. Cyanobacteria extract and vermicompost have phyto-stimulating characteristics, leading to increased plant growth and yield attributes in cotton plants. Cyanobacteria extract and vermicompost have phyto-protective activity since their components promote the defense response mechanisms in cotton plants cultivated in salt-affected soil and treated with longer irrigating intervals. Our results suggest that the integrated application of cyanobacteria extract and vermicompost represents a good, practically implementable approach that enables the utilization of low soil quality, especially in arid and semi-arid areas, as it obviously alleviates the adverse impacts of water stress in saline soil. Together, the use of cyanobacteria extract and vermicompost constitute a very favorable strategy for enhancing cotton growth and productivity and fiber quality under increasing irrigation intervals in salt-affected soil.

## Figures and Tables

**Figure 1 plants-12-01872-f001:**
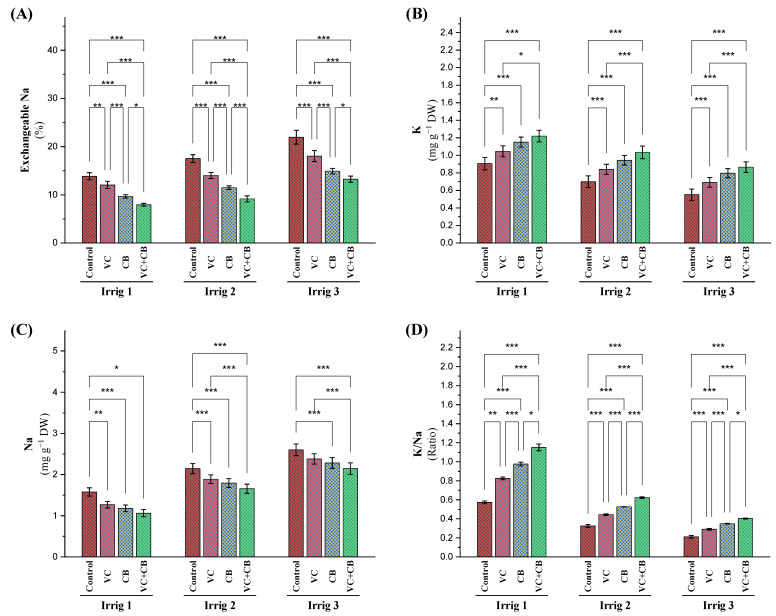
Influence of applying vermicompost (VC) and cyanobacteria extract (CB) on (**A**) exchangeable Na percentage (ESP) in the soil and the content of (**B**) sodium (Na) and (**C**) potassium (K) and (**D**) the K/Na ratio in the leaves of cotton plants grown in salt-affected soil and at three irrigation intervals (14, 21, and 28 days) during the 2020 and 2021 growing seasons. The data of both seasons are presented as means ± SD, and the significance levels of *, **, and *** indicate *p*-values of less than 0.05, 0.01, and 0.001, respectively.

**Figure 2 plants-12-01872-f002:**
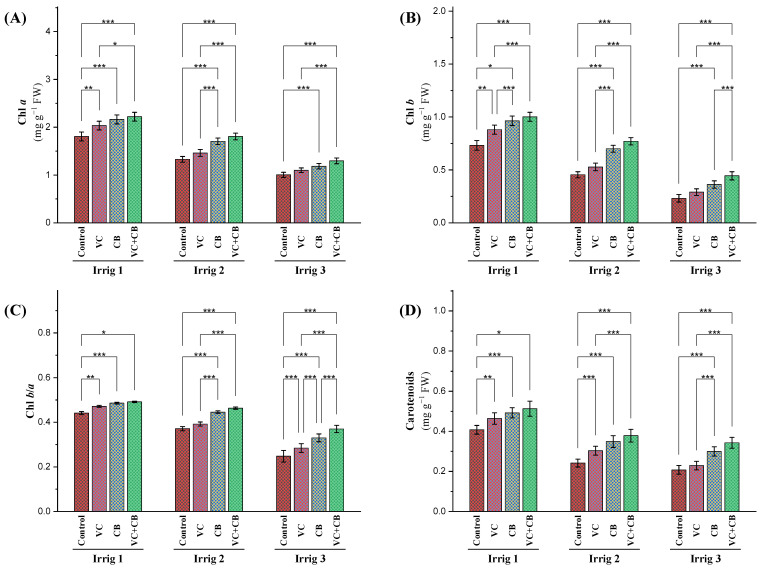
Influence of applying vermicompost (VC) and cyanobacteria extract (CB) on the concentration of (**A**) chlorophyll *a*, (**B**) chlorophyll *b*, (**C**) ratio between chlorophyll *b*/*a*, and (**D**) carotenoid content in the leaves of cotton plants grown in salt-affected soil and under three irrigation intervals (14, 21, and 28 days) during the 2020 and 2021 growing seasons. The data of both seasons are presented as means ± SD, and the significance levels of *, **, and *** indicate *p*-values of less than 0.05, 0.01, and 0.001, respectively.

**Figure 3 plants-12-01872-f003:**
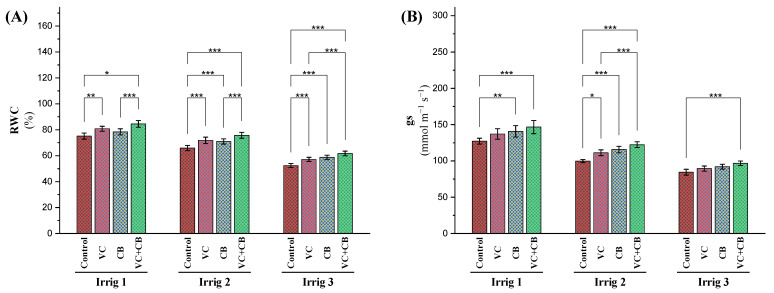
Influence of applying vermicompost (VC) and cyanobacteria extract (CB) on (**A**) relative water content (RWC) and (**B**) stomatal conductance (gs) in the leaves of cotton plants grown in salt-affected soil and under three irrigation intervals (14, 21, and 28 days) during the 2020 and 2021 growing seasons. The data of both seasons are presented as means ± SD, and the significance levels of *, **, and *** indicate *p*-values of less than 0.05, 0.01, and 0.001, respectively.

**Figure 4 plants-12-01872-f004:**
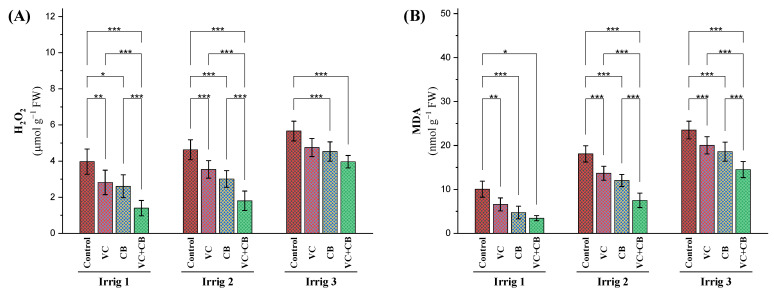
Influence of applying vermicompost (VC) and cyanobacteria extract (CB) on the stress indicators (**A**) hydrogen peroxide (H_2_O_2_) and (**B**) malondialdehyde (MDA) in the leaves of cotton plants grown in salt-affected soil and under three irrigation intervals (14, 21, and 28 days) during the 2020 and 2021 growing seasons. The data of both seasons are presented as means ± SD, and the significance levels of *, **, and *** indicate *p*-values of less than 0.05, 0.01, and 0.001, respectively.

**Figure 5 plants-12-01872-f005:**
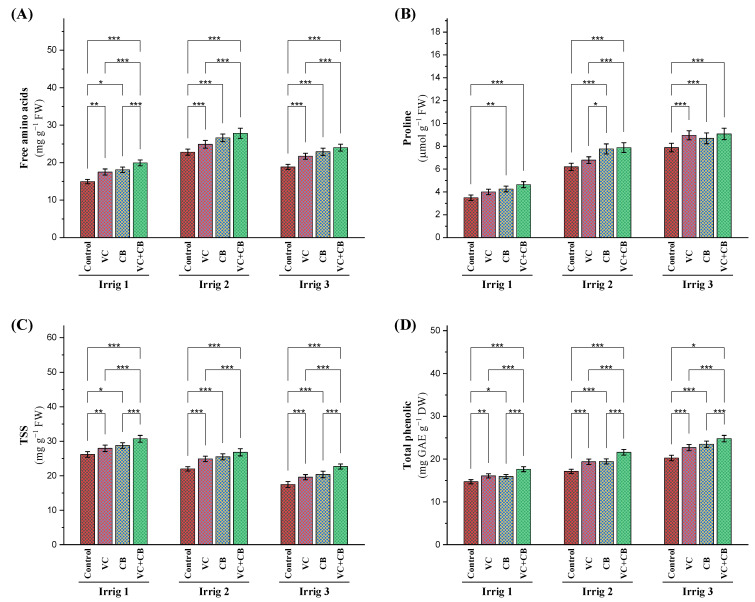
Influence of applying vermicompost (VC) and cyanobacteria extract (CB) on non-enzymatic antioxidants (**A**), free amino acids, (**B**) proline, (**C**) total soluble sugars, and (**D**) total phenolics in the leaves of cotton plants grown in salt-affected soil and under three irrigation intervals (14, 21, and 28 days) during the 2020 and 2021 growing seasons. The data of both seasons are presented as means ± SD, and the significance levels of *, **, and *** indicate *p*-values of less than 0.05, 0.01, and 0.001, respectively.

**Figure 6 plants-12-01872-f006:**
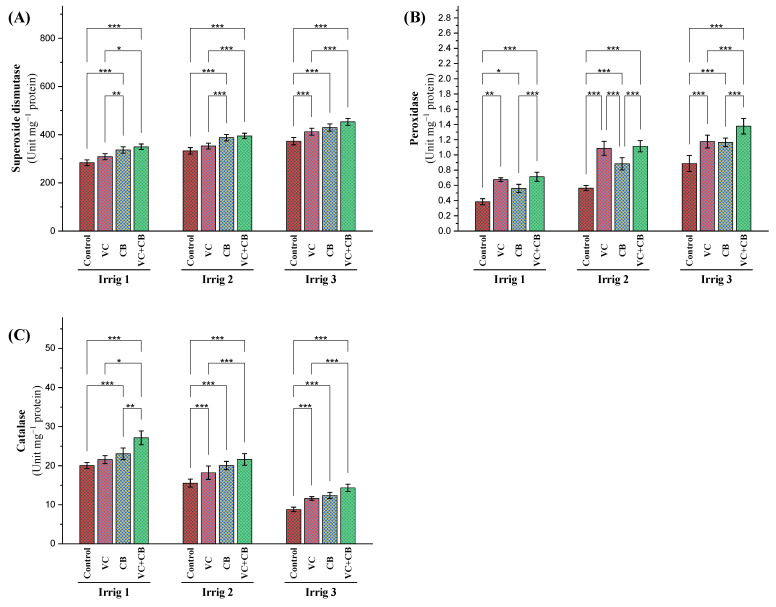
Influence of applying vermicompost (VC) and cyanobacteria extract (CB) on enzymatic antioxidants, namely, (**A**) superoxide dismutase, (**B**) peroxidase, and (**C**) catalase, in the leaves of cotton plants grown in salt-affected soil and under three irrigation intervals (14, 21, and 28 days) during the 2020 and 2021 growing seasons. The data of both seasons are presented as means ± SD, and the significance levels of *, **, and *** indicate *p*-values of less than 0.05, 0.01, and 0.001, respectively.

**Figure 7 plants-12-01872-f007:**
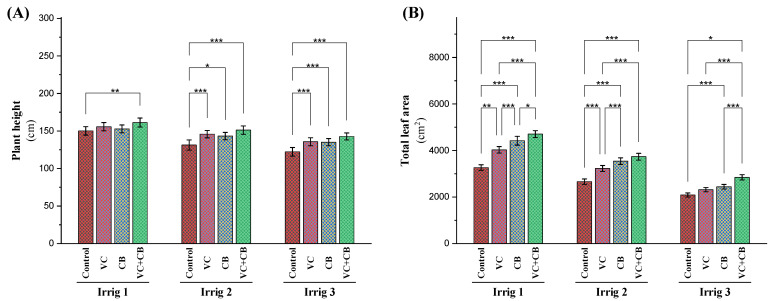
Influence of applying vermicompost (VC) and cyanobacteria extract (CB) on (**A**) plant height, and (**B**) total leaf area in cotton plants grown in salt-affected soil and under three irrigation intervals (14, 21, and 28 days) during the 2020 and 2021 growing seasons. The data of both seasons are presented as means ± SD, and the significance levels of *, **, and *** indicate *p*-values of less than 0.05, 0.01, and 0.001, respectively.

**Figure 8 plants-12-01872-f008:**
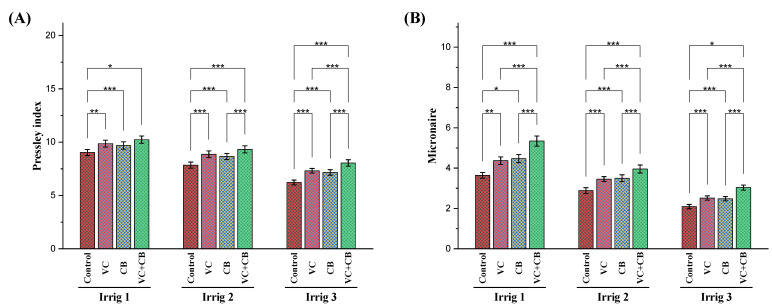
Influence of applying vermicompost (VC) and cyanobacteria extract (CB) on fiber quality parameters—(**A**) Pressley index and (**B**) Micronaire reading—on cotton plants grown in salt-affected soil and under three irrigation intervals (14, 21, and 28 days) during the 2020 and 2021 growing seasons. The data of both seasons are presented as means ± SD, and the significance levels of *, **, and *** indicate *p*-values of less than 0.05, 0.01, and 0.001, respectively.

**Table 1 plants-12-01872-t001:** Influence of applying vermicompost (VC) and cyanobacteria extract (CB) on yield characteristics of cotton plants grown in salt-affected soil and under three irrigation intervals (14, 21, and 28 days).

Irrigation	Treatments	No. of Fruiting Branches	Open Bolls No. Plant^−1^	Boll Weight (g)	Lint %	Seeds Yield (kg ha^−1^)
14 days	Control	10.4 ± 0.5 e	19.5 ± 0.6 e	2.74 ± 0.10 e	40.6 ± 1.3 bc	3057 ± 126.7 e
	Vermicompost	12.5 ± 0.6 c	22.4 ± 0.7 b	2.94 ± 0.09 b	41.7 ± 1.2 ab	3459 ± 125.8 b
	Cyanobacteria	13.0 ± 0.5 b	21.3 ± 0.7 c	2.86 ± 0.08 c	41.2 ± 1.3 b	3350 ± 112.7 c
	Combined	14.3 ± 0.6 a	23.8 ± 0.7 a	3.03 ± 0.10 a	42.9 ± 1.5 a	3627 ± 138.5 a
21 days	Control	8.5 ± 0.4 i	13.8 ± 0.5 i	2.55 ± 0.11 h	36.5 ± 1.1 f	2651 ± 97.3 i
	Vermicompost	9.9 ± 0.3 g	17.9 ± 0.6 f	2.67 ± 0.09 f	39.7 ± 1.3 cd	2922 ± 114.6 f
	Cyanobacteria	10.2 ± 0.3 f	16.6 ± 0.6 g	2.65 ± 0.08 f	39.1 ± 1.2 d	2853 ± 104.5 g
	Combined	10.8 ± 0.3 d	20.0 ± 0.7 d	2.77 ± 0.10 d	40.7 ± 1.4 bc	3118 ± 112.4 d
28 days	Control	6.0 ± 0.3 l	9.2 ± 0.3 l	2.32 ± 0.09 j	34.1 ± 1.8 g	2319 ± 108.2 k
	Vermicompost	8.0 ± 0.4 k	12.2 ± 0.5 j	2.58 ± 0.10 gh	37.7 ± 1.2 ef	2577 ± 98.1 j
	Cyanobacteria	8.2 ± 0.3 j	11.6 ± 0.5 k	2.51 ± 0.09 i	37.0 ± 0.7 f	2584 ± 100.6 j
	Combined	8.8 ± 0.3 h	15.0 ± 0.5 h	2.6 ± 0.12 g	38.6 ± 1.3 de	2746 ± 103.7 h
F-test	Irrigation	***	***	***	***	***
	Treatments	***	***	***	***	***
	Irrigation X Treatments	***	***	**	**	***

Means ± standard deviation followed by different lowercase letters indicate significant differences between treatments according to Tukey’s HSD Test (*p* < 0.05). The significance levels of ** and *** indicate *p*-values of less than 0.05, 0.01, and 0.001, respectively.

**Table 2 plants-12-01872-t002:** Meteorological data for the two growing seasons 2020 and 2021.

Year Month	2020				2021			
	Temperature (°C)		Wind Speed (km Day^−1^)	RH ^‡^ (%)	Temperature (°C)		Wind Speed (km Day^−1^)	RH (%)
	max ^¥^	min ^†^			max	min		
April	27.21	11.98	119.25	57.45	26.12	12.54	110.32	55.41
May	29.7	13.6	105.0	59.8	27.9	15.4	91.0	62.7
June	34.6	19.0	105.1	65.5	34.5	15.4	101.0	63.4
July	34.9	21.9	97.1	64.5	33.0	21.0	101.1	64.1
Aug.	34.5	19.8	79.5	63.4	35.0	22.2	91.5	66.9
Sept.	33.5	19.5	83.3	68.5	34.4	19.8	82.2	67.4
Average	33.4	18.1	101.9	67.3	32.9	17.4	93.3	66.1

^¥^ max = maximum, ^†^ min = minimum, ^‡^ RH = relative humidity.

**Table 3 plants-12-01872-t003:** Physicochemical characteristics of the experimental soil in the two growing seasons, 2020 and 2021.

Characteristics		2020	2021
pH (1:2.5 soil/water suspension)			
Soil depth (cm)	0–20	8.22 ± 0.02 ^†^	8.28 ± 0.03
	20–40	8.19 ± 0.02	8.21 ± 0.03
	40–60	8.16 ± 0.04	8.18 ± 0.02
Electrical conductivity (ECe, dS m^−1^) ^¥^			
Soil depth (cm)	0–20	5.61 ± 0.01	5.66 ± 0.02
	20–40	5.56 ± 0.02	5.59 ± 0.05
	40–60	5.36 ± 0.03	5.54 ± 0.04
ESP ^#^ (%)			
Soil depth (cm)	0–20	22.61 ± 0.42	21.50 ± 0.32
	20–40	22.52 ± 0.02	21.36 ± 0.22
	40–60	22.46 ± 0.04	21.21 ± 0.35
Soil organic matter (g kg^−1^)		11.2 ± 0.03	11.7 ± 0.05
*Particle size distribution* (*%*)			
Sand		27.22 ± 1.88	27.17 ± 1.98
Silt		25.23 ± 2.02	25.55 ± 1.99
Clay		47.55 ± 2.32	47.28 ± 2.03
Texture grade		clayey	clayey
*Soluble cations* (*meq L*^−1^) ^¥^			
Ca^++^		7.54 ± 0.94	9.29 ± 0.87
Mg^++^		5.76 ± 1.11	6.23 ± 1.32
Na^+^		26.75 ± 2.06	22.63 ± 3.08
K^+^		0.33 ± 0.02	0.39 ± 0.02
*Soluble anions* (*meq L*^−1^) ^¥^			
CO_3_^− −^		nd ^‡^	nd
HCO_3_^−^		4.61 ± 0.56	3.34 ± 0.68
Cl^−^		24.56 ± 1.11	18.21 ± 1.15
SO_4_^− −^		15.13 ± 3.03	11.15 ± 3.04
*Available macronutrients* (*mg kg*^−1^)			
N		9.70 ± 0.91	10.33 ± 1.71
P		8.24 ± 1.33	8.94 ± 1.54
K		344 ± 26.42	387 ± 24.33

^†^ Standard deviation; ^‡^ not detected; ^¥^ measured in soil paste extract; ^#^ exchangeable sodium percentage.

## Data Availability

Not applicable.

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
