# Peer review of "Mitigating Osmotic Stress and Enhancing Developmental Productivity Processes in Cotton through Integrative Use of Vermicompost and Cyanobacteria"

_plants, 2023, doi:10.3390/plants12091872_

Round 1
Reviewer 1 Report
I would like to thank you for your confidence in reviewing this manuscript.
I send you here my comments for the manuscript review.
Type of manuscript: Article
Title: Mitigating Osmotic Stress and Enhancing the Developmental Productivity Processes in Cotton through Integrative use of Vermicompost and Cyanobacteria.
Comments: Could authors provide some photos, which present the fields experiments and the different applied treatments?
Abstract:
Line 26: Please, put the signification of “CAT, SOD and POD” before their abbreviations.
Line 27: Please, put the signification of “H2O2 and MDA” before their abbreviations. All other abbreviations need to be explained before their use in the main text.
Line 35: Replace “stress” by “stresses”.
Introduction:
Line 42 and Line 49: Add a coma before “which”. Same remark in all the main text.
Line 44: Replace “zone” by “zones”.
Line 61: Replace “reduction photosynthesis process” by “reduction of photosynthesis process”.
The objective is missing.
Results:
Line 128: What is the meaning of “ESP”? Please, put it for the 1st time.
Figure 1:
Change the emplacement of Figure 1C with 1B. Figure 1C needs to 1B and 1B to 1 C.
Line 162: Add a coma before “respectively”. Same remark in all the main text.
Line 167: Replace “Figure 2” by “Figures 2A, B).
Line 205: “RWC” needs to be defined for the 1st time.
Line 279: Replace “Figure 5” by “Figures 5A, B).
Materials and methods:
Line 654 and Line 655: Replace “42 %” and “1.36 %” by “42%” and “1.36%”. Same remark in all the main text for %.
Line 748: Replace “4°C” by “4 °C”. Same remark in all the main text for °C.
Before conclusion, the Statistical Analysis needs to be incorporated.
References:
Please check references (in text and list) in relation to the journal's recommendations. The reference list needs to be revised carefully.
Author Response
Journal: Plants
Manuscript ID: plants-2332199
Type: Article
Number of Pages: 27
Title: Mitigating Osmotic Stress and Enhancing the Developmental Productivity Processes in
Cotton through Integrative use of Vermicompost and Cyanobacteria.
Authors: Khadiga Alharbi 1, Emad M. Hafez 2,*, Alaa El-Dein Omara 3, and Hany S. Osman4
Reviewer 1.
Could authors provide some photos, which present the fields experiments and the different applied treatments?
Response: Thank you so much for your kind comment. Our team doesn't have a plant breeder to focus on phenotype images. Most of us are plant and crop physiology experts, making us more focused on extracting results in numbers than presenting them as phenotype images. This is the first time we have been requested to add phenotype images to our papers, which we have not added before in our previously published articles in prestigious journals. We will take your point of view into our account in future research.
Abstract:
Line 26: Please, put the signification of “CAT, SOD and POD” before their abbreviations.
Response: We agree with the reviewer in line 26-27, done, thank you very much for your comment and corrected.
Line 27: Please, put the signification of “H2O2 and MDA” before their abbreviations. All other abbreviations need to be explained before their use in the main text.
Response: Done,line 30 this issue was double-checked, revised, and corrected where needed throughout the manuscript.
Line 35: Replace “stress” by “stresses”.
Response: Done, the abstract has been double-checked line 36
Introduction:
Line 42 and Line 49: Add a coma before “which”. Same remark in all the main text.
Response: DONE, this issue was double-checked. Done, thank you very much for your comment and corrected in lines 44 and 51..
Line 44: Replace “zone” by “zones”.
Response: Done, thank you very much for your comment and corrected in line 46.
Line 61: Replace “reduction photosynthesis process” by “reduction of photosynthesis process”.
Response: Thank you very much for your feedback, we totally agree with you. Changed in line 63.
The objective is missing.
Response: We would like to express our special gratitude to you for your insightful comment, we agree totally with you when we revised again carefully and we added the objective in lines 122-124
Results:
Line 128: What is the meaning of “ESP”? Please, put it for the 1st time.
Response: Done, thank you very much for your comment and corrected in line 142.
Figure 1:
Change the emplacement of Figure 1C with 1B. Figure 1C needs to 1B and 1B to 1 C.
Response:
Thank your for your comment. We presented figure 1B first followed by Figure 1C.
Line 162: Add a coma before “respectively”. Same remark in all the main text.
Response: Done, this issue was double-checked, revised, and corrected where needed throughout the manuscript.
Line 167: Replace “Figure 2” by “Figures 2A, B).
Response:Done, Thank your for your comment.
Done. Check line 190.
Line 205: “RWC” needs to be defined for the 1st time.
Response: Done, thank you very much for your comment and corrected in line 220.
Line 279: Replace “Figure 5” by “Figures 5A, B).
Response:
Done. Check line 303.
Materials and methods:
Line 654 and Line 655: Replace “42 %” and “1.36 %” by “42%” and “1.36%”. Same remark in all the main text for %.
Response: Done, thank you very much for your comment and corrected in line 685 and 686.
Line 748: Replace “4°C” by “4 °C”. Same remark in all the main text for °C.
Response: Done, thank you very much for your comment and corrected in line 778.
Before conclusion, the Statistical Analysis needs to be incorporated.
Response: Done, thank you very much for your comment and added in line 824-827.
References:
Please check references (in text and list) in relation to the journal's recommendations. The reference list needs to be revised carefully.
Response: Done, this issue was double-checked, revised, and corrected where needed throughout the manuscript.

Reviewer 2 Report
This manuscript was studied the effect of vermicompost and cyanobacteria extract on osmotic stress and developmental productivity processes of cotton in salt soils. The results concluded that the integrated application of vermicompost and cyanobacteria extract alleviated the osmotic stress and increased cotton productivity and fiber quality. I think the topic is of interest and better understanding of the roles of vermicompost and cyanobacteria in salt stress.
Several amendments are suggested as follows:
Lines 15-23, please concentrate these sentences.
There are too many descriptive text in the Introduction section, please focus on the advances of the effect of vermicompost and cyanobacteria on osmotic stress.
Lines 124-127, this paragraph can be deleted or placed into the end of the Introduction section as the objective of the study.
Line 128, please explain the “ESP”.
The symbols of irrigation treatment (Irrig 1, 2, 3) in the figures should be consistent with that (Irrig. 14, 21, 28) in the text.
The Discussion should be improved. It should be based on the results.
Author Response
Journal: Plants
Manuscript ID: plants-2332199
Type: Article
Number of Pages: 27
Title: Mitigating Osmotic Stress and Enhancing the Developmental Productivity Processes in Cotton through Integrative use of Vermicompost and Cyanobacteria.
Authors: Khadiga Alharbi 1, Emad M. Hafez 2,*, Alaa El-Dein Omara 3, and Hany S. Osman4
Reviewer 2.
This manuscript was studied the effect of vermicompost and cyanobacteria extract on osmotic stress and developmental productivity processes of cotton in salt soils. The results concluded that the integrated application of vermicompost and cyanobacteria extract alleviated the osmotic stress and increased cotton productivity and fiber quality. I think the topic is of interest and better understanding of the roles of vermicompost and cyanobacteria in salt stress.
Response: Firstly, thank you very much for your nice words. We appreciate all the reviewer’s comments and suggestions, which definitely enhanced the quality of the manuscript.
Several amendments are suggested as follows:
Lines 15-23, please concentrate these sentences.
Response: Done, this issue was double-checked, revised, and corrected
There are too many descriptive text in the Introduction section, please focus on the advances of the effect of vermicompost and cyanobacteria on osmotic stress.
Response: We really respect your constructive criticism that can improve and strengthen the research before it is published. Done, this issue was double-checked, revised, and added in lines 94-98.
Lines 124-127, this paragraph can be deleted or placed into the end of the Introduction section as the objective of the study.
Response: Done, thank you very much for your comment, we deleted it and placed into the end of the Introduction section as the objective of the study
Line 128, please explain the “ESP”.
Response: Done in line 134, thank you very much for your comment
The symbols of irrigation treatment (Irrig 1, 2, 3) in the figures should be consistent with that (Irrig. 14, 21, 28) in the text.
Response:
Thank you for your comment. In our opinion, the figures should be standalone units that are self-explanatory. Therefore, we relied on the full description of irrigation treatments in the figure caption and used a simpler representation of the irrigation treatments within the figure itself to avoid having to write out lengthy descriptions within the figure (such as "irrigation every 14 days") and to avoid repetition.
The Discussion should be improved. It should be based on the results.
Response: Thank you so much for your kind comment, we really respect your constructive criticism that can improve and strengthen the research before it is published. We have already divided the discussion into sub-sections as indicated in the discussion section to be more clarity.

Round 2
Reviewer 1 Report
Non comment
Reviewer 2 Report
The manuscript was revised according to the reviewers' comments and suggestions. I think this version can be accepted for publication in the journal.